# Unexpected large eruptions from buoyant magma bodies within viscoelastic crust

Freysteinn Sigmundsson [1✉], Virginie Pinel [2], Ronni Grapenthin [3], Andrew Hooper [4], Sæmundur A. Halldórsson[1], Páll Einarsson [1], Benedikt G. Ófeigsson[5], Elías R. Heimisson [6], Kristín Jónsdóttir[5], Magnús T. Gudmundsson [1], Kristín Vogfjörd[5], Michelle Parks[5], Siqi Li [1], Vincent Drouin [7], Halldór Geirsson [1], Stéphanie Dumont [8], Hildur M. Fridriksdottir[5], Gunnar B. Gudmundsson[5], Tim J. Wright [4] & Tadashi Yamasaki [9]

Large volume effusive eruptions with relatively minor observed precursory signals are at odds with widely used models to interpret volcano deformation. Here we propose a new modelling framework that resolves this discrepancy by accounting for magma buoyancy, viscoelastic crustal properties, and sustained magma channels. At low magma accumulation rates, the stability of deep magma bodies is governed by the magma-host rock density contrast and the magma body thickness. During eruptions, inelastic processes including magma mush erosion and thermal effects, can form a sustained channel that supports magma flow, driven by the pressure difference between the magma body and surface vents. At failure onset, it may be difficult to forecast the final eruption volume; pressure in a magma body may drop well below the lithostatic load, create under-pressure and initiate a caldera collapse, despite only modest precursors.

[1] Nordic Volcanological Center, Institute of Earth Sciences, University of Iceland, IS-101 Reykjavik, Iceland. [2] Univ. Grenoble Alpes, Univ. Savoie Mont Blanc, CNRS, IRD, IFSTTAR, ISTerre, 38000 Grenoble, France. [3] Geophysical Institute & Dept. of Geosciences, University of Alaska Fairbanks, 2156 Koyukuk Drive, Fairbanks AK-99775, USA. [4] COMET, School of Earth and Environment, University of Leeds, Leeds, UK. [5] Icelandic Meteorological Office, Reykjavik, Iceland. [6] Seismological Laboratory, California Institute of Technology, Pasadena, CA, USA. [7] Iceland GeoSurvey, Reykjavik, Iceland. [8] Instituto Dom Luiz - University of Beira Interior, Covilhã, Portugal. [9] Geological Survey of Japan, AIST, 1-1-1 Higashi, Tsukuba, Ibaraki 305-8567, Japan. ✉email: fs@hi.is

Some of the largest effusive eruptions on Earth in the last 10,000 years were basaltic fissure eruptions in Iceland, with volumes ranging from 15–25 km³ (ref. [1–3]). More recently, during the 2014–2015 Holuhraun eruption ~2 km³ of magma were removed from a magma body[4,5]. Yet, precursory signals prior to magma body failure were weak, which raises questions about how magma bodies supporting such large eruptions are assembled, how eruptions are initiated and how large volumes of magma are extracted without major precursors.

The 1783–1784 Laki and the 2014–2015 Holuhraun fissure eruptions in Iceland respectively produced 15 km³ and 1.4 km³ of basaltic lava, formed by concurrent mixing and crystallisation of diverse primary mantle-derived melts[6,7]. Such moderate to large effusive basaltic eruptions and the preceding long-term magma accumulation, modulated by tectonics and mantle plume activity, provide a challenge for commonly used mechanical volcano models. This is clearly illustrated by the 2014–2015 activity in the Bárðarbunga volcanic system, when gradual caldera collapse occurred over a period of 6 months in response to drainage of

magma from below the caldera, initially along a lateral dike[5,8] that fed a major effusive eruption 45 km away at Holuhraun (Supplementary Note 1). Several lines of evidence (ground deformation, petrology and seismic activity)[4,5] suggest that the magma came from a single magma body located at a depth of 10 ± 3 km beneath the surface of the ice-filled caldera, below the brittle-ductile transition[4,7,9]. Although deformation and seismicity suggest some magma inflow prior to the eruption (Fig. 1, Supplementary Note 1, Supplementary Fig. 1), the amount of deformation suggests the short-term pre-eruptive magma inflow volume was minor compared to the volume drained during the eruption, unless the magma was highly compressible, damping observable inflation[10]. This is unlikely given the storage conditions and magma composition. This implies that overpressure (pressure in excess of lithostatic pressure) due to magma inflow at the onset of the 2014–2015 events was small compared to the co-eruptive pressure drop; a feature not in line with commonly used mechanical volcano models.

The host rock surrounding a magma body is typically modelled as elastic, responding instantaneously to a pressure change in a

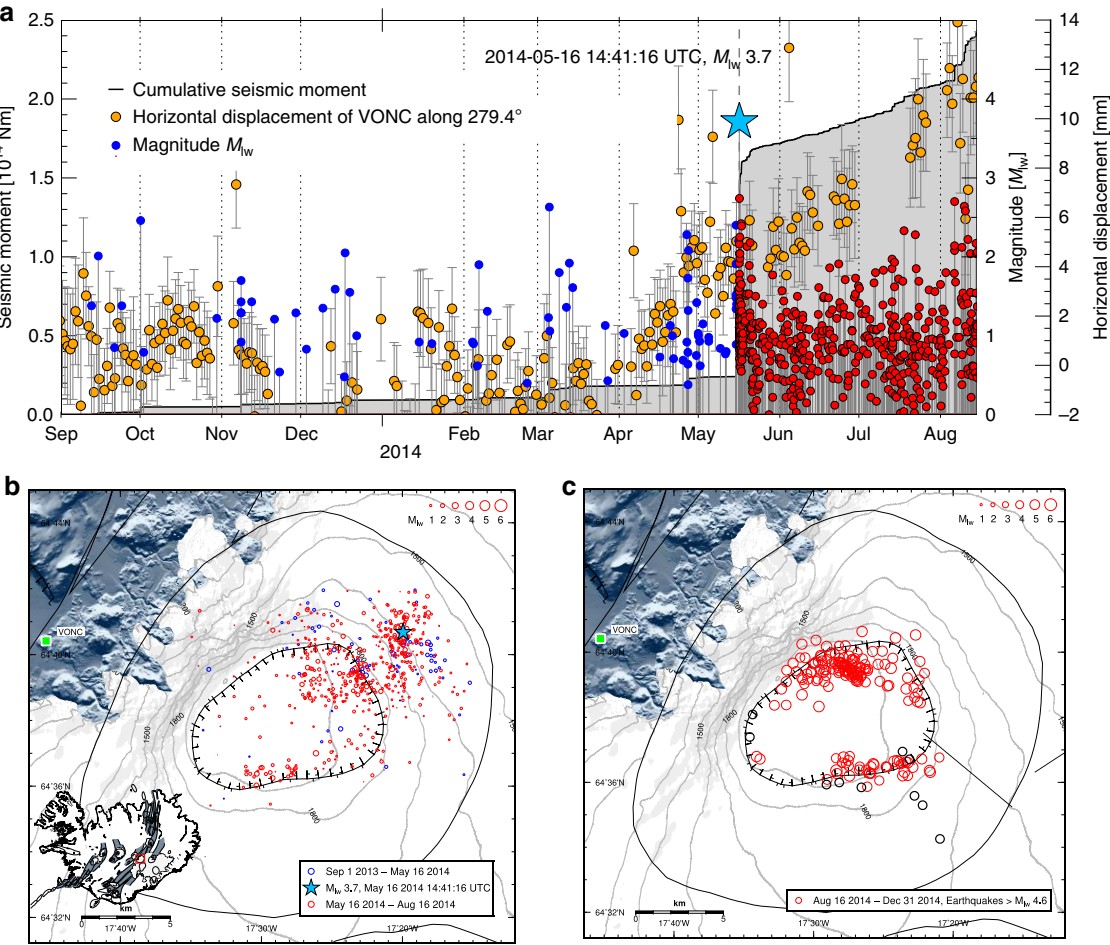

**Fig. 1 Seismicity and deformation at Bárðarbunga. a** Earthquakes versus time (1 September 2013–15 August 2014; before diking and eruption) plotted as impulses scaled with magnitude (right axis). Earthquakes prior to M3.7 event on 16 May 2014 shown in blue and red afterwards. Also shown is cumulative seismic moment (shaded in grey; left axis), and horizontal displacement in direction N279.4°E (yellow dots) at GPS-station VONC from detrended time series (Supplementary Fig. 1). Error bars (1σ) in grey. **b** Inferred location of the earthquakes (Methods) shown in **a**, with earthquakes prior to M3.7 event on 16 May 2014 shown in blue and red afterwards. Small map of Iceland shows the study area outlined with a red box, fissure swarms[76] with grey shading, and oval outlines showing central volcanoes and calderas[76]. **c** Location of M > 4.6 earthquakes during the caldera collapse (Methods). Note aseismic segments of the caldera. Background map shows shaded surface (grey) and Vatnajökull icecap (white) topography from the ArcticDEM database[77], and outlines of the Bárðarbunga central volcano (oval shape) and its caldera[42]. Straight lines show segments of the lateral dike that formed and black open circles are ice cauldrons[5].

source dependent on geometry and elastic stiffness. The density difference between magma and host rock is usually also neglected. In such models, the erupted volume can only be a small fraction of total magma chamber volume[11,12]. As a result, the generation of large eruptive volumes would require very large magma reservoirs, high magma compressibility, or high overpressure that may far exceed the effective tensile strength of the Earth. High overpressure is, however, expected to induce large pre-eruptive surface displacements, not observed at, for instance, Bárðarbunga. Furthermore, in elastic models neglecting buoyancy effects, an eruption should stop before any large depressurisation of a magma reservoir can be reached thus preventing the formation of a caldera[13,14]. By contrast, in models where the Earth rheology is viscoelastic, a pressure change induces an initial elastic response followed by ductile relaxation, with relaxation rates that depend on the viscosity of the medium. While long-lived magma bodies may develop a local thermal aureole, modelled as a pressure source with a surrounding viscoelastic shell[15,16], magma is often stored in multiple sills on its way towards the surface, with progressive compositional evolution[17,18]. For deep bodies, embedded within sufficiently hot media, we can expect entire crustal sections to behave in a viscoelastic manner[19]. Such long-term ductile behaviour of the Earth also gives rise to glacial isostatic adjustment (GIA) (ref. [20]).

While viscoelasticity is important to consider over the long timescales of magmatic recharge, over the short timescales associated with eruption and brittle failure, the host rock can be approximated as elastic. Furthermore, buoyancy of magma is important to consider when modelling eruption onset, as it has been suggested to play a key role in initiating super-eruptions[21]. Pressurisation associated with magma injection, on the other hand, is thought to be responsible for relatively small and frequent eruptions[21].

Here we consider the consequences of buoyant magma accumulating in a viscoelastic medium, followed by magma draining through sustained magma conduits that allow magma to flow, despite development of under-pressure (pressure less than lithostatic) in the magma body. We develop a physical model that explains this paradox, which is not addressed by widely used volcano deformation models[22] that only incorporate elastic crustal behaviour without considering density contrasts between magma and crust. But if magma accumulation occurs over a long timescale compared to the viscoelastic relaxation time of the host rock, and the magma is less dense than the host rock, the dominant factor driving magma bodies towards failure is magma buoyancy. Our model can be used to determine the conditions under which caldera collapse may initiate without significant precursors and drive large volumes of magma from depth to the surface. The model can account for the sequence of events that led to the 2014-2015 Bárðarbunga caldera collapse[4] and its associated unrest and eruption[5].

## Results

**Modelling framework.** When magma injects into a new or pre-existing magma body embedded in viscoelastic host rock, characterised by a linear viscoelastic rheology (Maxwell material), the stress relaxation timescale ($\tau$) is

$$\tau = \gamma \frac{\eta}{\mu}. \qquad (1)$$

Here the ratio of host rock viscosity ($\eta$) and shear modulus ($\mu$) is the so-called Maxwell time, and $\gamma$ is a factor dependent on both the geometry of the magma body and the viscoelastic structure[15,16,19,23]. In general, the viscosity is highly temperature-dependent and the uppermost few kilometres of the crust behave effectively as an elastic material. Appropriate regional values for

the lower crust in Iceland are $\eta = (2–8) \times 10^{18}$ Pa s, as inferred from GIA in response to present day ice thinning[20], and $\mu = 10$–30 GPa (ref. [24]), giving a regional Maxwell time on the order of 2–25 years (for $\gamma = 1$). While host rock viscosity at active volcanic plumbing systems is expected to be drastically lower than average regional values, due to higher temperature, $\gamma$ is on the order of 10 or higher for magmatic sills emplaced in a viscoelastic medium under an elastic layer[19]. Considering these competing effects, the relaxation time of host rock next to sill-like bodies due to magma inflow may be comparable to, or shorter than, the regional Maxwell time inferred from GIA; short compared to the time of magma accumulation over inter-eruptive periods lasting decades to centuries. At low emplacement rates, it follows that viscoelastic host rock rheology allows large magma volumes to accumulate over long periods without developing large overpressure, as stresses generated around a magma body due to inflow of new magma are relaxed.

In a viscoelastic material the density difference between magma and crustal rocks, $\Delta\rho$, generates a buoyancy force that is directed vertically upward for a magma body as a whole (Figs. 2 and 3), with magnitude given by Archimedes law. At the boundary of the magma body, the buoyancy pressure is locally $\Delta\rho g h$, where $g$ is acceleration of gravity, and $h$ is the height above the base of the magma body (Methods). Above a magma body, the resulting buoyancy-induced stresses are similar to those created by magma inflow (Fig. 4). Pressure at the boundary of a magma body results from the combined effects of buoyancy and pressure changes arising from magma inflow or outflow ($\Delta P_{\text{magma body,flow}}$).

A failure criterion for a buoyant magma body of uniform density, impacted by external stress, can be written as:

$$\Delta\rho g h + \Delta P_{\text{magma body,flow}} \geq \sigma_{\text{failure}} + \sigma_{\text{external}} \qquad (2)$$

where $\sigma_{\text{failure}}$ is the failure limit. The external stress ($\sigma_{\text{external}}$; negative for tension) is the component of deviatoric stress at the boundary of the magma body due to all external processes acting perpendicular to the plane of failure. This can result from a combination of processes, including tectonic, short- to long-term surface loading such as GIA, and topographic effects. We refer to Eq. (2) as the general failure criterion for a magma body. Failure is approached when either (i) a buoyant magma body increases in thickness, (ii) its density contrast increases through magma evolution, (iii) its pressure increases due to influx of magma, (iv) external stresses bring the deviatoric state of the host rock closer to failure, or (v) a combination of any of these. If the rupture is governed by tensile failure[11] and in agreement with magma-filled dikes and hydrofracture mechanical models[25], then $\sigma_{\text{failure}}$ is independent of depth and equals the effective large-scale tensile strength ($T_{\text{eff}}$) multiplied by a geometrical factor $\alpha$ ($\sigma_{\text{failure}} = \alpha T_{\text{eff}}$). The geometrical factor $\alpha$ depends on the magma body geometry and takes a value close to 1 or less for a horizontally elongated shape[12]. The effective large-scale tensile strength ($T_{\text{eff}}$) is on the order of few MPa (refs. [26,27]), up to an order of magnitude lower than laboratory values for intact rocks[28]. Alternatively, if failure occurs by shear faulting, as sometimes proposed in numerical gravity-loaded elasto-plastic models, the failure limit is up to an order of magnitude higher (several tens of MPa) (refs. [29,30]). Buoyancy effects limit the maximum magma body thickness. For example, in the case of tensile failure, if $\Delta\rho = 270$ kg/m³, $T_{\text{eff}} = 2.5$ MPa and in the case $\alpha = 1$ and $\sigma_{\text{external}} = 0$, the thickness has to be <1 km (Fig. 2) in order to ensure stability. The tectonic setting is also important; at divergent plate boundaries the stress field supports less internal pressure accumulation before failure of the host rock surrounding the magma body than at convergent margins.

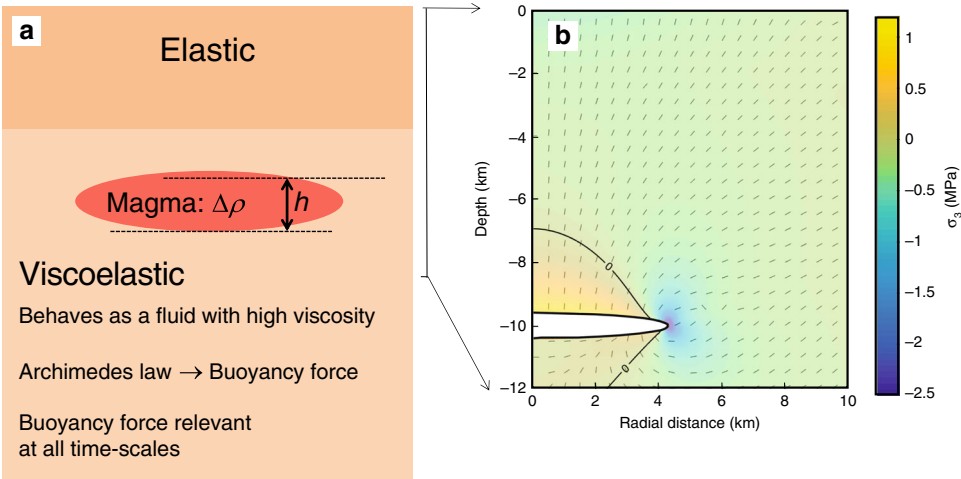

**Fig. 2 Model and stress field due to buoyancy. a** Schematic model of a magma body residing in viscoelastic material below an elastic layer. The magma is less dense than the host rock and therefore buoyant. The density difference between magma and host rock is $\Delta\rho$, and $h$ is the elevation above the bottom of the magma body. **b** Stress due to buoyancy around a horizontal ellipsoid, with a 4.3-km semi-major axis and a 0.4-km semi-minor axis, embedded in an elastic medium with a Poisson's ratio of 0.25. The density contrast is set to 270 kg/m$^3$. Pressure equal to $\Delta\rho gh$, with $g$ being the acceleration of gravity, is applied at its boundary (Fig. 3). The colour scale shows the amplitude of the minimum compressive stress (negative values for tension) and dashes represent the direction of maximum compressive stress ($\sigma_1$). The crustal volume immediately above the magma body is under compression, but the edge of the body is affected by a large tensile stress allowing for dike initiation. Above the body, the stress field is similar to the one induced by overpressure resulting from magma inflow (Fig. 4).

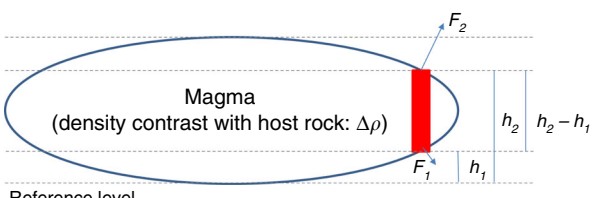

**Fig. 3 Buoyancy modelled as traction on boundary of a magma body.** Overpressure is calculated with reference to the deepest point of the magma body. This pressure is transferred to a surface force at the boundary, in a direction normal to it. The vertically directed upward force on the column shown in red is $\Delta A \Delta\rho g$ ($h_2 - h_1$). Here $\Delta A$ is a horizontal cross-sectional area element, equal to $(\cos\varphi)\delta A$, where $\varphi$ is the local angle that the boundary of the magma body makes with respect to horizontal. The integrated upward force due to all columns in the magma body gives an overall upward force following Archimedes law. For this geometry, the integrated horizontal force on the entire boundary will be zero (cancels out for this geometry because of symmetry).

Following magma body failure, a channel will form enabling magma outflow. Its shape depends on several processes and factors, such as tectonic and topographic stresses acting on the channel[31], zones of weaknesses such as caldera bounding faults, and the buoyancy pressure[32]. If the channel has vertical extent from depth $D_1$ to depth $D_2$, then the driving pressure ($\Delta P_{driving}$) for upward flow in the channel can be written[33]:

$$\Delta P_{driving} = \Delta\rho gh + \Delta P_{magmabody, flow} + \int_{D_1}^{D_2} \Delta\rho g dz. \quad (3)$$

The first two terms are internal to the magma body and the same as in Eq. (2): the magma body buoyancy, and the effects of magma flow to and from it. The third term captures effects of magma buoyancy in the channel. If crustal and magma density are constant with depth, then it simplifies to $\Delta\rho gH$, where $H$ is the height of the channel equal to $D_2 - D_1$. Equation (3) applies to

magma flow in a channel only after the establishment of a stable geometry and before its final phase of closure, when flux decreases and the magma cools and solidifies. We refer to this as a sustained channel. Within it, upward magma flow will continue as long as the driving pressure remains positive. At the channel top, the overpressure compared to lithostatic pressure (Fig. 5) equals the magma flow driving pressure, neglecting effects of magma velocity and friction (Methods). Near a deflating magma body, the channel may experience high compressive stress resulting in its closure. Exceptions may include situations influenced by inelastic processes. One such process is addition of magma mush (consisting of both crystals and some interstitial liquid) to the magmatic liquid flowing out of a magma body[7,34], evidenced in some cases by the presence of mineral assemblages, including macrocrysts, that are not in equilibrium with the host lava (Methods). A continuous entrainment process is inferred if the residence time of macrocrysts in the magma, prior to eruption, is short compared to the duration of eruption, and macrocrysts are relatively uniformly distributed in lava, as found for example, in the 2014–2015 Holuhraun lava[7,35]. Relatively uniform content of macrocrysts, despite variable magma flow rates during an eruption, indicates furthermore that the amount of material added to the lava can scale with the magma flow rate. We refer to this process as "magma mush erosion", which generates space for a magma channel via a process not included in elastic models, supporting the formation of a sustained channel. Thermal effects may also contribute: solidification of thin parts of a magma-filled crack due to cooling along the host rock interface will help to keep open the wider parts of a dike through bridging effects. Conversely, the wider parts of a dike may grow even wider through thermal erosion. This is the same process by which eruptive fissures that initially produce "curtains of fire" eruptions evolve with time; some parts of fissures close while others may widen and focus activity[36]. Tectonic effects, e.g., the release of tectonic stress in extensional environments may also contribute to the formation of sustained magma channels.

The deep roots of volcanoes may host a complex system of sills and dikes forming transcrustal magmatic systems with variable amounts of magma mush[18,34]. This can be referred to as magma

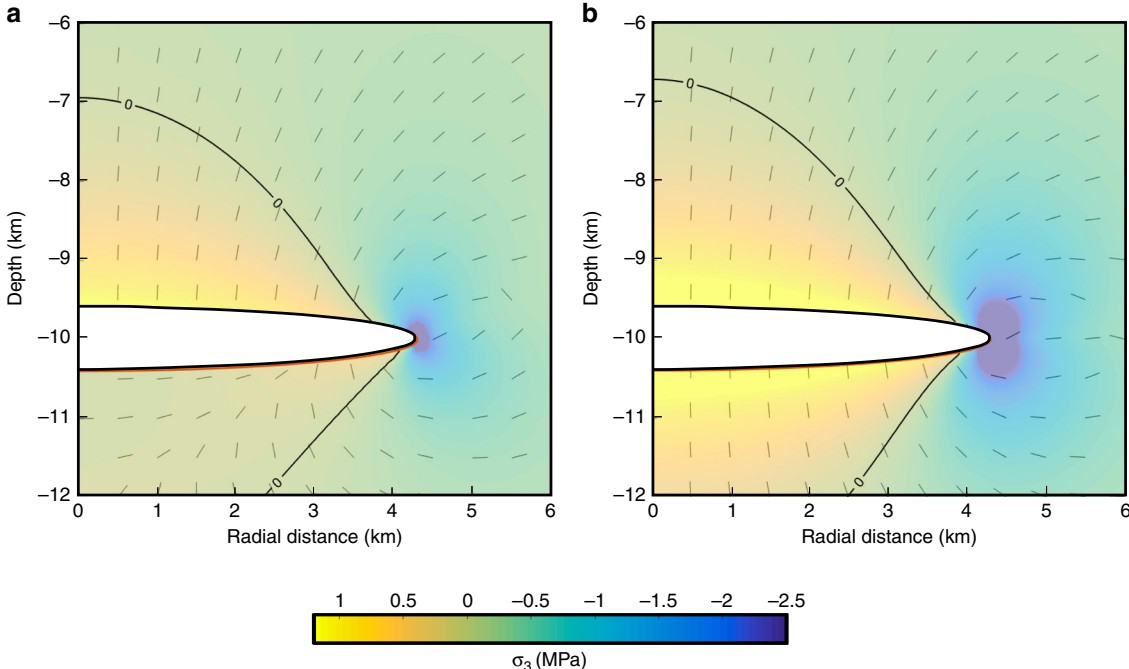

**Fig. 4 Comparison of stress field at magma bodies due to buoyancy and magma flow. a** Zoom of Fig. 2b showing stress field set up due to buoyancy of magma body. **b** Stress field due to 2.5 MPa pressure increase caused by magma inflow, for comparison. The colour scale shows the amplitude of the minimum compressive stress (negative values for tension) and the dashes represent the direction of maximum compressive stress ($\sigma_1$).

domain[37,38] and during an eruption magma extraction may occur from one or more magma bodies within the magma domain. In the instance of magma draining from a single magma body, its size and shape will provide a strong control on the magma flow. Calderas are often several kilometres or more in diameter, implying similar dimensions of underlying magma bodies. When magma is buoyant, the general failure criterion (Eq. 2) requires these bodies to be thin compared to their lateral extent if tensile failure applies. An oblate horizontal ellipsoid represents an appropriate geometric model. On a short timescale, the relation between the volume of magma outflow, $\Delta V_{magma}$, and the pressure change within an oblate ellipsoid, filled with a compressible fluid and embedded in an elastic halfspace with a Poisson's ratio of 0.25 and shear modulus $\mu$ is (Methods):

$$\Delta V_{\text{magma}} = \left(\frac{2}{\mu} + \frac{4\pi}{3}\frac{b}{a}\frac{1}{k}\right)a^3 \Delta P_{\text{magmabody, flow}} \tag{4}$$

where $1/k$ is the magma compressibility, $a$ is the semi-major axis of the ellipsoid and $b$ is its semi-minor axis. The equation shows that for an oblate ellipsoid with thickness much smaller than lateral extent, the effects of magma compressibility may be much smaller than for a more spherical magma body where the semi-minor axis approaches the length of the semi-major axis, as the compressibility is scaled by $b/a$. After failure, and with the formation of a sustained magma channel from the source, the pressure reduction can be much larger than the initial over-pressure required for failure (Fig. 5).

Volatile content and the chemical composition of magma strongly modulate rheological properties of magma (including its viscosity and density), and in turn, significantly influence subsurface magma flow and eruption dynamics[10,39]. For basaltic magma, the most important volatiles to consider are $H_2O$, $CO_2$ and S. For example, estimates of depth-dependent changes in magma density require information on magma composition and dissolved volatile content, in addition to an appropriate set of thermodynamic relationships describing how volatile distribution between gas and melt changes as pressure varies[40]. Given how magma density, $\rho_{magma}$, changes with pressure, the magma compressibility, $1/k$, can be inferred, as it equals $(1/\rho_{magma})(\partial\rho_{magma}/\partial p)$ where $p$ is pressure. Silica content and concentration of volatiles have thus also a large influence on the compressibility of magma[10,39]. In order to constrain the density difference between magma and the host rock, and thus infer the magma dynamics, a model of the crustal density profile is required.

**Application to Bárðarbunga volcano**. To resolve the discrepancy between minor precursory deformation and large volume eruption, we test our model against the 2014–2015 activity in the Bárðarbunga volcanic system. During this activity, moderately evolved olivine tholeiite magma is inferred to have drained from a single magma body located somewhere in the range of 7–13 km depth below the surface of the ice cover of Bárðarbunga[4,7,9,35]. In particular, geobarometry suggests that the most probable pressure at which the magma resided was $230 \pm 140$ MPa (refs. [7,35]). Crystal-hosted melt inclusions in the erupted lava[41] indicate that the magma contained about 0.5 wt.% $H_2O$, 500 ppm $CO_2$ and 1600 ppm S at 230 MPa prior to the eruption. We used two software programs to infer the magma density (Methods); one of which (the D-Compress software[40]) is used to infer variation in magma density and compressibility versus depth resulting from volatile loss (Fig. 6). A generalised model for the crustal density profile in Iceland[42] is modified to account for the ~1.8-km elevation of Bárðarbunga above sea level, divided into a 0.6-km-thick ice layer and 1.2-km-thick low-density hyaloclastite layer (Supplementary Text and Supplementary Fig. 2). The model gives crustal density versus depth and allows the determination of crustal pressure conditions (Fig. 6). Over the 150–230-MPa pressure range (~6–9 km depth below surface; ~4–7 km depth below sea level) the magma and crustal densities show minor variations; their difference being in the range of 270–300 kg/m³. For conservative modelling of the buoyancy effect, we initially take the lower limit of this range ($\Delta\rho = 270$ kg/m³). This density

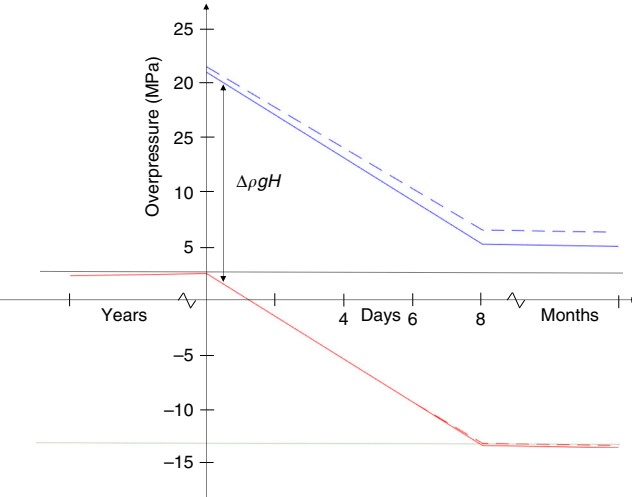

**Fig. 5 Excess pressure in a magma plumbing system.** Pressure history during inflation and deflation. Combined overpressure from magma flow and buoyancy pressure in a magma body (red, solid line considering a constant magma density and dashed line taking into consideration the magma density changes induced by gas exsolution) and in a location higher within a sustained magma channel (blue, solid line considering a constant magma density and dashed line taking into consideration the magma density changes induced by gas exsolution). The two curves are separated by $\Delta\rho g H$ (which is a constant value when volatiles are not considered). Note that volatiles are not expected to influence the overpressure evolution for a magma body located at 8 km or larger depth. Gas exsolution may cause the two curves to slightly diverge as deflation evolves, as such a process will always have a larger influence at shallower depth. Failure of the magma body occurs at $t = 0$ and caldera collapse (failure of caldera faults) begins in this example at day 8 after the failure of the magma body. For a long time prior to that (years), the pressure conditions are close to the magma body failure limit due to magma buoyancy. Overpressure in the magma body is limited according to the general failure criterion, here considered to correspond to be effective tensile strength of 2.5 MPa (black horizontal dotted line). In this example, slip on caldera ring faults begins at day 8 when critical conditions for caldera failure are reached (green horizontal line). Following that, there is small change in pressure as piston collapse drives out a large volume of magma over months. See text for discussion.

difference is also considered in the crustal layers next to the magma body to constrain the magma channel dynamics.

Considering a short relaxation timescale and the location of the magma body within a viscoelastic domain, stresses due to magma accumulation prior to the events may have been continuously relaxed over the 150 years since the previous major Bárðarbunga fissure eruption[1]. The Gjálp eruption in 1996 did, however, significantly modify conditions at Bárðarbunga. It was preceded by a 22-year period of ~15 magnitude 5+ earthquakes, the last one leading to a lateral dike propagating from Bárðarbunga. This 1996 dike intersected another magma body outside of the caldera[43], which provided the majority of erupted material. Ring fault activity occurred during this event, suggesting weak caldera faults. We infer that in August 2014 the general failure criterion (Eq. 2) was satisfied again at Bárðarbunga, resulting in buoyant magma draining from the underlying magma body (Fig. 7).

Contraction of a magma body under the caldera has been modelled[44] as a closing sill with lateral dimensions similar to the caldera. We consider a similar but simplified model geometry. We utilise a reference magma body and Earth model in our calculations, and then assess deviations from these. The reference magma body model is an oblate ellipsoid magma body at 10 km

depth below the surface with semi-major axis $a = 4.3$ km, so that the horizontal width of the ellipsoid is comparable to the dimensions of the Bárðarbunga caldera. We define the semi-minor axis such that tensile stress at the edge of the magma body does not exceed ~2.5 MPa due to buoyancy. This requires the maximum thickness of the magma body to be ~1 km or less. Here we set the semi-minor axis to be $b = 0.4$ km (Fig. 2). We set the compressibility ($1/k$) of the magma in the body to 0.1 GPa$^{-1}$, similar to the value derived from D-Compress at the depth of the magma body (0.09 GPa$^{-1}$). The effective Young's modulus of the Icelandic crust has been estimated ($40 \pm 15$) GPa based on annual cycles in ground displacements due to snow and ice load changes[24]. The corresponding shear modulus is $\mu = E/(2 + 2v) = (10-22)$ GPa. For the reference model we use the lower limit, $\mu = 10$ GPa, to reflect the expectation that crustal rigidity under volcanoes and in a magma domain may be lower than regional averages. Caldera collapse began on day 8 of the activity, heralded by the onset of M5+ earthquakes at the caldera[4]. According to ground deformation modelling[5], ~0.3 km$^3$ of magma had already flowed into the dike at that time. For this volume change and the parameter values above, the associated pressure change in the magma body, $\Delta P_{\text{magma body,flow}}$, is $-15.8$ MPa (Fig. 5).

Next, we consider a model for a magma channel that links the magma body under Bárðarbunga and the lateral dike that formed (Fig. 7). During the dike emplacement and eruption, elastic processes dominated, except for inelastic processes that may have created a sustained channel where Eq. (3) applies. Even if the entrained material from the crystal mush is only 1% of the drained magma volume, as we estimate for the 2014–2015 magma (Methods), then the volume of entrained magma mush would still be $\sim 20 \times 10^6$ m$^3$, corresponding to, for example, a 5-km high conduit segment (dike) with a $4 \times 10^3$ m$^2$ cross-sectional area (e.g., 4 km long and 1 m wide). We suggest a vertical upflow path along the eastern caldera boundary of Bárðarbunga (Supplementary Text). This initial flow path from the magma body may have been a dike that formed along a part of the caldera boundary where it was weak. At a higher level, within the brittle crust, where the magma is less buoyant and topographic and tectonic stress have major influence, a regional dike formed (Fig. 7). The maximum opening of the regional dike is at ~3 km depth (ref. [5]) and we consider this to be the average depth to which magma needs to rise in our reference model. If magma flowed vertically upward from 10 km to 3 km depth below the surface (approximate depth of maximum opening in the dike[5]), the overpressure gain in the vertical channel ($\Delta\rho gh$) is 18.5 MPa for $\Delta\rho = 270$ kg/m$^3$. This shows that the driving pressure can remain positive, allowing for flow despite the large pressure drop in the magma body (Fig. 5). Depressurisation in the model produces strong shear stress at the locations of potential ring faults at depth, which favours the initiation of a piston-type collapse (Fig. 8); large under-pressure allowed for the onset of movements on already weak caldera faults following previous events (Supplementary Text). Once the collapse was initiated, release of gravitational energy, associated with the subsiding piston[4], drove out a large volume of magma along the previously created magma path. During the piston collapse, exponential decay in magma flow-rate occurred with an estimated pressure change of <2 MPa, comparable to the change in lithostatic pressure as the caldera subsided 66 m (ref. [4]).

Although the values of the model parameters presented can explain the behaviour of the volcano in the framework of our model, their exact values are uncertain. If the magma – host rock density difference is in the range of 250–300 kg/m$^3$ and the depth extent of the sustained channel is in the range of 4–10 km (the average rise of magma from the magma body to the effective dike inlet), then the difference in pressure between the magma body and the effective dike inlet is in the range of 12–30 MPa (the

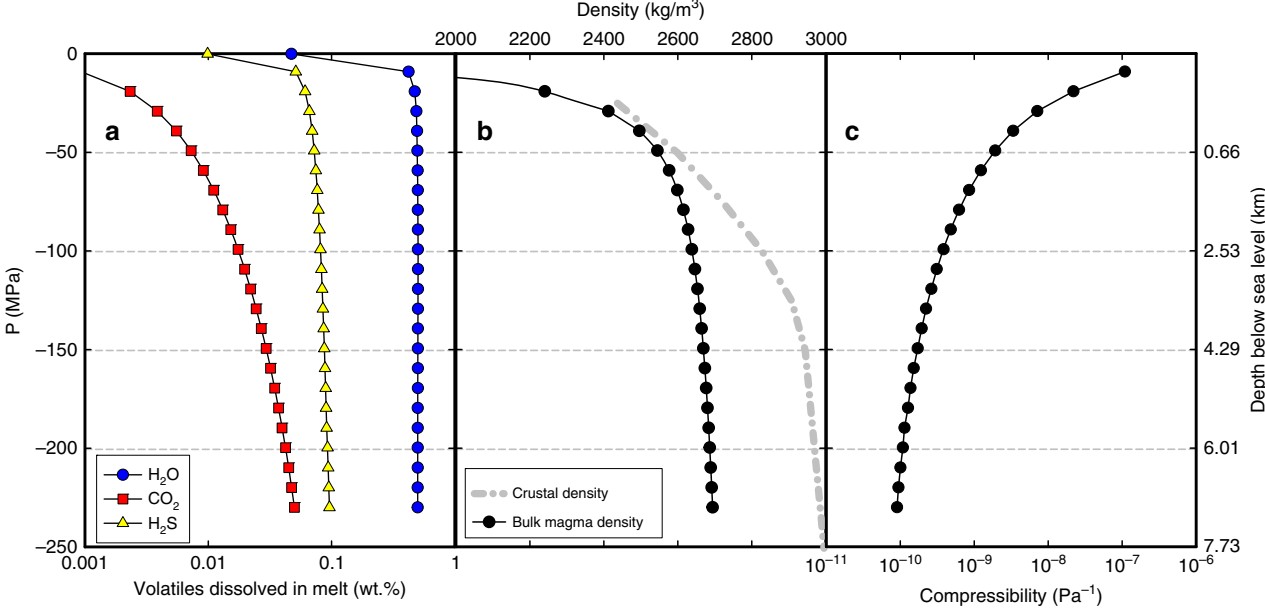

**Fig. 6 Volatile content, magma and crustal properties versus pressure/depth. a** Dissolved volatile contents (in wt.%) of the 2014–2015 Bárðarbunga magma versus pressure (in MPa, left vertical axis). **b** Bulk magma (black) and crustal (grey hatched) density (in kg/m$^3$) and **c** bulk magma compressibility (in Pa$^{-1}$). Conversion from pressure to depth below sea level is shown at the right vertical axis. See also Supplementary text and Supplementary Fig. 2.

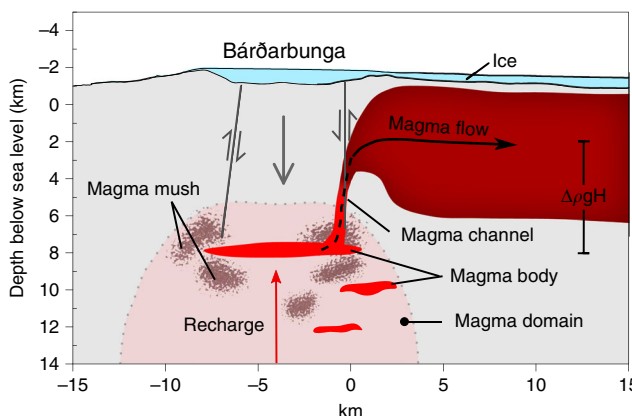

**Fig. 7 Schematic cross-section of a model for the 2014–2015 Bárðarbunga collapse.** A sill-like magma body resides within a magma domain[37,38] under Bárðarbunga. Modest inflow of magma (recharge) occurred prior to August 2014, at least in the last few months prior to activity (see Fig. 1). Failure of the magma body occurred because of the combined effects of buoyancy and pressure due to magma flow. After failure, magma flowed up from an exit at the magma body towards an effective inlet to a regional lateral dike at a higher level in the crust. Macrocrysts were entrained into the magma via erosion of a magma mush, contributing to the formation of a sustained magma channel. The magma channel may have been in the form of a dike along the caldera boundary (perpendicular to the plane of the figure). At shallower levels tectonic and topographic stresses dominated and a regional dike formed. Despite development of under-pressure in the magma body, the driving pressure for flow remained positive. The magma body involved was located beneath the long-term brittle-ductile transition located in the 6–8-km depth range below sea level, near the lower edge of the dike formed[9]. The structure of the magma domain is shown in a schematic manner. It may have other isolated magma bodies as shown and volumes of crystal mush; at least one volume of magma mush was involved in creating the erupted magma.

offset of the upper set of curves in Fig. 5 relative to the lower set). For the low end of this range the eruption may stop early because the driving pressure for the flow (Eq. 3) falls quickly to zero. If the amount of combined overpressure due to buoyancy and magma flow at the time of failure was higher than the suggested value of 2.5 MPa, then the pressure curves shown in Fig. 5 would all be shifted to higher levels. In turn, the amount of pressure drop needed to create under-pressure and onset of caldera collapse would be larger. The gradient of the two sets of the pressure curves in the time between the onset of magma outflow and when the caldera collapse begins, depends on all the parameters in Eq. (4). The larger the dimensions of the magma body, and the larger the compressibility, the lower the gradient of pressure drop versus time. However, for an oblate ellipsoid and the model parameters presented above, the dominant factor is the first term in the bracket in Eq. (4), which depends on the crustal shear modulus. The higher the shear modulus, the larger the pressure drop for a fixed magma outflow volume.

## Discussion

The model presented here explains how a magma body can develop an under-pressure that is an order of magnitude higher than the overpressure prior to initial failure, and yet facilitate substantial magma outflow, as we have demonstrated for the case of the 2014–2015 events at Bárðarbunga. This is different from the behaviour of a fully elastic model that does not consider gravity and where stresses never drop below those prior to the onset of magma accumulation, limiting the pressure drop to be equal to or less than the pre-eruptive pressure increase. In our model, a sustained conduit along part of a caldera boundary allows under-pressure to develop in the underlying magma body and, at the same time, reduces the overall resistance on the caldera boundary where it forms. Both factors facilitate a caldera collapse. The under-pressure that develops depends on the size of the magma body: the smaller the body the larger the under-pressure that develops for a fixed outflow volume. The development of under-pressure hinders upward flow of magma and may eventually stop it, but at the same time it increases the likelihood

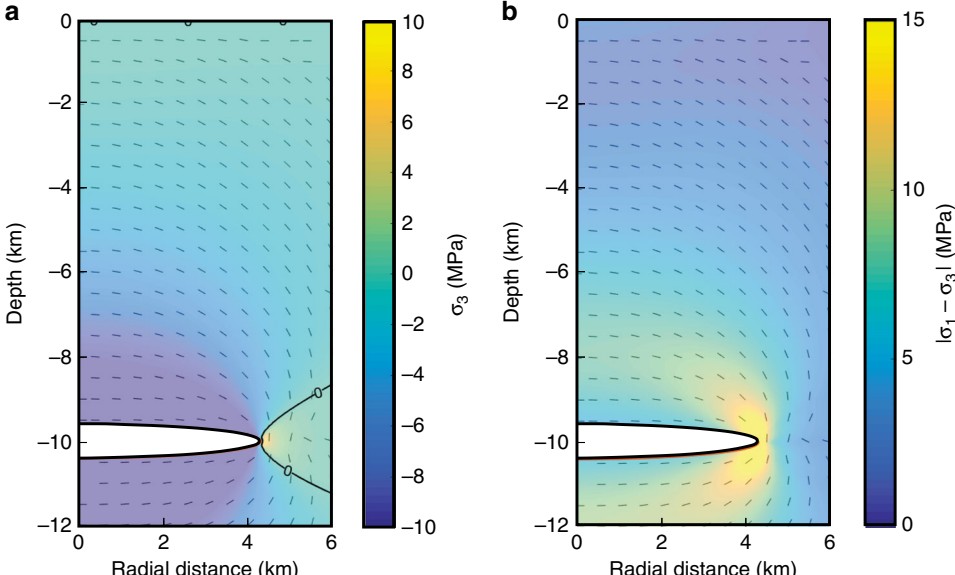

**Fig. 8 Stress fields related to major drainage of magma. a** Amplitude of the minimum compressive stress, shown by the colour scale with negative values for tension, induced by combined buoyancy and under-pressure due to magma outflow ($\Delta\rho gh + \Delta P_{magma\ body,flow}$) equal to −13.3 MPa, corresponding to the model presented for Bárðarbunga. Direction of maximum compressive stress ($\sigma_1$) is represented by dashes. Most of the medium is put under tension with maximum effect immediately above the magma body. **b** Shear stress (amplitude of $\sigma_1$–$\sigma_3$; shown by the colour scale) for the same conditions as in **a**. The panels show that the stress field is favourable for caldera collapse initiation induced by the depressurisation.

of a caldera collapse. Thus, the right conditions for initiation of a caldera collapse may rarely be met, consistent with their infrequent occurrence.

For water-poor basaltic magmas, exsolution of volatile elements from melts stored at mid- to deep-crustal levels (>8 km) has less of an effect on magma compressibility relative to water-rich, silicic magmas stored at shallow levels[10]. In addition, the presence of high-density fluid inclusions in the Holuhraun melt suggest that it became $CO_2$ saturated at mid- to lower-crustal pressures in excess of 400 MPa (ref. [41]). A considerable portion of the primary $CO_2$ was therefore lost via open-system degassing before the Holuhraun magma accumulated in the inferred magma body at ~230 MPa pressure (ref. [41]). This is consistent with other studies[45,46], indicating $CO_2$ exsolution from basaltic melt under Iceland and its subsequent loss begins at ~25 km depth. Our depth-dependent volatile exsolution modelling (Fig. 6) further suggests that the loss of $CO_2$ from the melt was a near-continuous process, while both water (which profoundly affects magma compressibility) and sulfur exsolution only become significant at shallow depths (<500 m). As a result, changes in rheological properties of the Holuhraun magma driven by volatile loss mostly occurred in the topmost three kilometres of the crust, above the level of maximum dike opening. In this depth range, volatile loss has thus a major influence on magma ascent dynamics whereas at deeper levels the effects on dike opening are minor.

Our model is appropriate for deep magma accumulation, with magma flow rates below a threshold value (Methods). For Bárðarbunga, the model provides a direct link between the deep magma body, where accumulation and drainage occurred, and the shallower regional dike. Another example of well-studied deep magma accumulation in extensional tectonics is the Socorro Magma Body in the Rio Grande Rift. It is inferred to be ~150 m thick, residing at ~19 km depth, and has been inflating at low rates for decades[47–49]. One would expect similar processes to be at work there, even if the lithospheric properties are different from those in Iceland. Individual elements of the modelling framework presented here (viscoelastic host rock behaviour, magma

buoyancy, sustained magma channel, development of under-pressure to initiate caldera collapse) may help to understand large volume eruptions in general. The most recent large volume effusive eruptions on Earth include the rift eruption at Kilauea volcano in 2018 when ~0.8 km³ of magma erupted in relation to summit caldera collapse[38,50], and the ongoing submarine eruption east of the Mayotte island, Indian Ocean, which began in May 2018 and where more than 5 km³ of magma have erupted[51–53]. At Kilauea, interpretation of geodetic measurements finds pressure drop prior to onset of caldera collapse to be ~17 MPa (ref. [50]), similar to what we find for the Bárðarbunga collapse. In the activity east of Mayotte, drainage from a deep magma body (>20 km depth) in the mantle is inferred[51,53] where the host rock is viscoelastic. Studies of the eruptive products show the magma was buoyant[52]. More general, the reawakening of dormant volcanoes by inflow and accumulation of magma to shallow depth may result from failure of deep-seated buoyant magma bodies in viscoelastic host rock, residing in the lower crust or the mantle. If magma buoyancy plays a role in bringing magma bodies towards the general failure criterion, then our model suggests failure of such bodies can occur with little warning, and once initiated, can result in unanticipated large lava production.

## Methods

**Location of earthquakes.** Earthquakes shown in Fig. 1 are from routine locations from the national seismic network of Iceland operated by the Icelandic Meteorological Office, shifted southward[4], to correct for inadequacy of the velocity model that does not account for lateral variations in velocity structure. The southward shift decreases linearly from ~2.6 km at the northern caldera rim to 0.5 km at the southern rim. This shift is applied both to earthquakes in the pre-eruptive period and the diking/eruptive period shown in Fig. 1b, c, respectively. Formal location errors were significantly improved in second half of August 2014 as six new seismic stations were deployed within 45 km radius of the caldera (seven stations before). After late August 2014 the formal epicentral location errors are typically up to 1 km, whereas before they are up to 5 km.

**Buoyancy pressure.** The effect of magma buoyancy on dynamics of spherical magma bodies has been considered before[21,54–56], but is here expanded to magma bodies of arbitrary shape. The total upward force acting on a magma body of lower density than the surrounding viscoelastic crust is found, according to Archimedes

law, by considering the integrated body force due to gravity acting on the magma body minus the gravity force of the crustal material that the magma body has replaced. At the magma body boundary this upward directed body force is transferred to a surface force acting on the surrounding crust. Figures 2 and 3 show the case for an ellipsoidal magma body. For the case of uniform density contrast, the total upward force that the magma body exerts on the surrounding viscoelastic crust is $\Delta\rho g V$, where $\Delta\rho$ is the difference in density between magma and crust, $g$ the gravitational acceleration, and $V$ the volume of the magma body. For variable density, $\Delta\rho$ is the difference of the average densities of the magma and the crust. As the magma body is embedded in a material that behaves as a viscous fluid on a long timescale, it will have the tendency to "float" upwards and its shape may be distorted. Stokes flow for bodies emerged in viscous fluids, and its extension to a fluid body emplaced within another fluid body suggest that upward movement velocity[57,58] will be on the order of $(1/3)\beta\Delta\rho g a^2/\eta$, where $a$ is the semi-major axis for an ellipsoidal magma body and $\beta$ is a scaling factor that takes into consideration the aspect ratio $b/a$ ($b$ being the semi-minor axis of the magma body). If $a = b$ (spherical body), then $\beta = 1$. If $b < a$ (ellipsoid) then $\beta$ is <1. Assuming such a body within the Icelandic lower crust with viscosity of $5 \times 10^{18}$ Pa s, density difference $\Delta\rho$ equal to 270 kg/m$^3$, and $a = 4.3$ km, the upward rate of movement can be expected to be <10 cm/yr. This is slow enough that a magma body receiving inflow from depth is likely to transfer magma from deeper levels to shallower levels via feeder dikes, rather than diapiric rise of whole sills through the crust. Using the same values, the growth rate of Rayleigh-Taylor instabilities is expected to remain smaller than a few millimetres per century[59], confirming that the ascent of magma caused by buoyancy will be effective only when the magma body thickness will be large enough to overcome the surrounding medium resistance to rupture.

**Viscous pressure drop induced by magma flow**. Equation 3 gives the pressure at the top of the vertical magma channel neglecting the viscous pressure drop induced by the magma flow. This pressure drop can be estimated[60] around $\eta_m v H/w^2$, where $\eta_m$ is the magma viscosity, $v$ the magma velocity and $w$ the channel width. Taking $\eta = 100$ Pa s, $w = 1$ m, $v = 1$ m/s and $H = 7$ km, the viscous pressure drop is ~0.7 MPa or <4% of the buoyancy gain over the channel height (18.5 MPa; see main body of text). The overestimation of the pressure at the horizontal dike inlet resulting from neglecting the magma flow is thus expected to be small.

**Sustained magma channel**. Mantle-derived lavas containing plagioclase and other mineral assemblages, too primitive to be in equilibrium with the host lava, are widely distributed around the globe[3,61]. One of the most striking examples of such crystal-melt disequilibria are so called plagioclase-ultra-phyric basalts which are found within the Neovolcanic zone and throughout the near 16 Ma lava pile of Iceland[62]. These lavas are characterised by an abundance (often >10 vol%) of large crystals (>~0.5 mm; macrocrystals) with high relative proportion of plagioclase to olivine. A widespread view is that these macrocrystals crystallised at middle to deep crustal levels and accumulated in a crystal mush layer from which they were picked up by their host/carrier melt during ascent. Well documented examples of Icelandic Holocene lavas, where prior studies have demonstrated the importance of such a magma mush layer, include the 8600 BP Þjórsárhraun lava[3], the 1783 Laki lava[6,63], and more recently the 2014–15 Holuhraun lava[7,35]. Consideration of elemental diffusion in these crystals and their melt inclusions demonstrates that timescales associated with the crystal entrainment process are likely to be short, or equivalent to timescales of these eruptions, i.e., a few weeks to months[3,7,64]. This suggests that the crystals have been brought into the magma only shortly before or during these eruptions. In line with these observations, we suggest that an important process for formation of sustained magma channels is the addition of mush crystals to a magmatic liquid as it leaves a magma body and rises towards the surface. In this case, a near-continuous addition of mush crystals into the host/carrier melt[63], in proportion to the rate of magma flow, leads to uniform crystal content throughout the erupted magma, despite variable rate of flow. We suggest that erosion of such a crystal mush layer may ultimately lead to formation of sustained openings, which are required for the extraction of a melt from a magma body. The reduction in volumetric stress which happens above and below a deflating sill (Fig. 8) may indeed make the crystal mush more easily erodible and therefore accommodate this process.

For the 2014–2015 Holuhraun lava, the amount of macrocrysts, defined as minerals with long axes >1 mm in hand specimen, is relatively small compared to some older Bárðarbunga lavas. Halldórsson et al.[35] and later work estimated the range to be from ~1% and up to no more than 5% in terms of volume. Yet, even the lower bound of 1% is sufficient to support the role of magma mush erosion in the formation of a sustained channel in the 2014-2015 events.

**Horizontal oblate ellipsoid filled with compressible fluid**. The volume change of a penny-shaped crack of radius $a$, embedded in an elastic halfspace with shear modulus $\mu$ and Poisson's ratio 0.25, in response to a pressure change $\Delta P$ is[65]

$$\Delta V_{penny} = \frac{2a^3}{\mu}\Delta P. \tag{5}$$

The volume change of a fluid body in response to pressure change due to its compressibility is $V\Delta P/k$, where $1/k$ is the fluid compressibility. If the fluid is in the

form of an ellipsoid with semi-major axis $a$ and semi-minor axis $b$, then its volume is $(4/3)\pi a^2 b$. In the case of magma outflow from an ellipsoidal body that takes the form of a penny-shaped crack ($b \ll a$), the volume of magma outflow equals the combined volume change of a penny-shaped crack and the volume change due to compressibility. Adding these two terms together gives Eq. (4).

**Numerical modelling**. The conceptual model presented here relies on the ductile behaviour of the lower crust surrounding a magma body. Nevertheless, we use numerical models considering elastic rheology to support it. This approach is justified since the buoyancy force in a viscoelastic medium, contrary to the overpressure induced by magma inflow, cannot be relaxed through time by a change in volume or shape. Its effect can be quantified using an elastic medium and taking the reference state of stress as lithostatic (fully relaxed). The stress field resulting from a pressurised buoyant body is thus determined from the equations for linear elasticity which we solved using the Finite Element Method in axisymmetrical geometry with the COMSOL software. We use a mesh of about 120,000 triangular units that is refined around the magma body (maximum element size of 20 m next to the magma body). No displacement is allowed at the lateral and bottom boundaries, which are set at a distance equal to 20 times the depth of the magma body whereas the upper boundary is considered a free surface. Stress due to buoyancy is similar to the one induced by an overpressurized magma body (Fig. 4). When the combined stress exceeds a given threshold, it favours tensile rupture of the medium consistent with magma propagation through the medium. Within a ductile medium, tensile stress is expected to favour rupture and magma propagation through sheet intrusions the same way as in an elastic medium except for a difference of the tip shape, the tip being less sharp in a ductile medium[66].

**Density of magma**. The density of the 2014–15 Holuhraun magma was calculated using both the D-Compress as well as the Petrolog3 software[67,68]. In the latter case, we assumed that whole-rock composition of the lava flow represents the melt compositions. From the average of 62 whole-rock analyses[35], a density value of $2722 \pm 5$ kg/m$^3$ was obtained for the 2014–15 Holuhraun melt at 230 MPa (corresponding to 2.3 kbar). Pressure dependency of these calculations is negligible and within the error introduced by considering standard deviation of the mean value of 62 whole-rock analyses adopted for the calculations. A fixed Fe$^{3+}$/Fe$^{total}$ ratio of 0.15 was assumed, which is considered a representative value for evolved rift-associated basalts in Iceland. Water content of 0.5 wt% was selected as it best characterises prior-degassing $H_2O$ levels of the Holuhraun magma at ~230 MPa. The inferred density value is in good agreement with values obtained for aphyric lavas of similar compositions and volume from North Iceland[45].

**Density of the crust at Bárðarbunga**. The Icelandic crust is 3–4 times thicker than normal oceanic crust[69]. However, the upper crust in Iceland is similar in structure to that of the normal oceanic crust, with high gradients in both seismic P-wave velocities and density in the uppmost 5 km, while the gradients are much lower at greater depth[70]. Two density models for basaltic crust, based on empirical data, are those of Carlson and Herrig[71] and Christensen and Wilkens[72]. The latter model is based on velocity-density systematics from a 1.9-km deep drillhole in east Iceland. These models have been used to constrain the sources of gravity anomalies in the upper crust in the Bárðarbunga region[42]. We use these models together with ice cover information to obtain an estimate of the density-lithostatic pressure systematics for Bárðarbunga. The Carlson and Herrig model provides a minimum density estimate and the Christensen and Wilkens model a maximum. We use the mean of these two models to determine the most plausible pressure-density function (Supplementary Fig. 2). The ice layer is assumed to be 600-m thick and to have a density of 900 kg m$^{-3}$. The density in the uppermost kilometre of the crust has considerable uncertainty. We use a mean value of 2300 kg m$^{-3}$ for the rocks at the ice-bedrock interface and a strong gradient down to 4 km depth below sea level (density 2950 kg m$^{-3}$, pressure of ~130 MPa, see Supplementary Fig. 2). The lower panel in Supplementary Fig. 2 shows density as a function of depth. Depth values refer to Bárðarbunga, where zero is sea level.

**Volatiles and magma properties versus depth**. Following Kilbride et al.[10], we adopted the D-Compress thermodynamic model of volatile saturation and partitioning[40] to predict the composition of a gas phase in equilibrium with the 2014–15 Holuhraun melt as a function of volatile content, melt composition, temperature, pressure and oxidation state as the magma rose from the magma body at ~7.5 km depth b.s.l. (230 MPa) to the surface (based on the crustal density model; Supplementary Fig. 2). On the basis of these output parameters, we calculated magma compressibility considering the exsolved gas phase following $(1/\rho_{magma})(\partial\rho_{magma}/\partial p)$ where $\rho$ is density and $p$ is pressure. Assuming an oxygen fugacity buffer of NNO+0.5, a temperature of 1170 °C and pressure of 230 MPa (ref. [35]), our modelling demonstrates that $CO_2$ will continue to be effectively lost from melt (Fig. 6a), reaching a near-zero (~0.001 wt.%) value in erupted and fully degassed melt at 0.1 MPa (1 bar), consistent with observations from the Holuhraun glass[41]. However, once vapour saturation is reached, sulfur (we only plot melt-dissolved $H_2S$ in Fig. 6a) exsolves continuously, with a major loss restricted to pressures below 10 MPa. Moreover, water remains largely dissolved in the melt until pressures of about 5 MPa (<200 m). Rapid and efficient exsolution and resulting bubble

growth is therefore expected to occur once the melt has reached such low pressures. To quantitatively estimate the effects on magma buoyancy at mid-crustal levels, it is important to consider the resulting change in magma density. At 230 MPa, D-Compress calculates a bulk density of the magma of 2694 kg/m$^3$, almost identical to the value calculated for melt (2698 kg/m$^3$) at the same depth. These values are somewhat lower than the density inferred from the whole-rock composition of the lava flow described above (2722 kg/m$^3$). D-Compress is used to calculate the density (Fig. 6b) and compressibility (Fig. 6c) changes of the bulk magma as a function of depth.

**Threshold for basal inflow rate**. Magma may accumulate in a deep magma body without inducing its rupture if the mean magma basal inflow rate, $\phi$, is small enough such that the volume of magma entering the body over a time period equal to the relaxation time ($\Delta V = \phi\tau$) does not induce pressure increase beyond the rock tensile strength. Using Eq. (4), we can thus derive an upper threshold value for the mean basal inflow:

$$\phi < \left(2 + \frac{4\pi}{3}\frac{b}{a}\frac{\mu}{k}\right)\frac{a^3 T}{\tau\mu}. \tag{6}$$

Taking $a = 4.3$ km, $b = 0.4$ km, $T = 2.5$ MPa, $\mu = k = 10$ GPa, and the relaxation time as 10 years, the threshold value for the mean basal flow would be about 5 million m$^3$/yr. This value is smaller than the inferred rate of basal inflow in the nearby Grímsvötn volcano[73,74] in the period from 2004 to 2012, consistent with the higher frequency of eruption observed at Grimsvötn[75]. Considering this threshold value for the mean basal flow, ~6000 years would be required to accumulate the 30 km$^3$ of magma in an ellipsoid of dimensions suggested in this example, which is an approximate estimate of the maximum size of the magma body that fed the 2014–2015 activity under Bárðarbunga.

## Data availability
The source data underlying Figs. 1, 2b, 4, 6, and 8 and Supplementary Figs. 1 and 2 are provided in a Source Data file.

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

## Acknowledgements

The research presented here has benefitted from extended visits of FS during a sabbatical term to, and discussion with scientists at, the University of Leeds, ISTerre University of Savoie Mont-Blanc, USGS Cascades Volcano Observatory, and Geological Survey of Japan. We acknowledge reviews by Philip Benson and Luca Caricchi that helped to significantly improve the paper, as well as reviews of an early version of the paper by two anonymous reviewers. Financial support from the H2020 project EUROVOLC funded by the European Commission is acknowledged (grant number 731070). F.S. acknowledges support from the University of Iceland Research Fund, and R.G. acknowledges partial support through NSF grant EAR-1464546. Fissure swarms, central volcanoes and caldera outlines shown in Fig. 1 are reproduced from publications referred to (refs. [42,76]) with permissions from Elsevier, and we acknowledge the use of ArticDEM (ref. [77]) to plot surface and ice topography shown in Fig. 1. COMET is the NERC Centre for the Observation and Modelling of Earthquakes, Volcanoes and Tectonics, a partnership between UK Universities and the British Geological Survey.

## Author contributions

F.S. led the development of the ideas and modelling framework presented in this paper, with the participation of V.P., R.G., A.H., S.A.H., P.E., E.R.H., M.T.G., T.W. and T.Y.; Numerical modelling with the COMSOL software was carried out by V.P., and S.A.H. did the D-Compress modelling. The crustal density model was made by M.T.G.; Collection and analyses of seismic and geodetic data was carried out by B.G.Ó., K.J., K.V., M.P., S.D. H.M.F., G.B.G, P.E., E.R.H, H.G., F.S., A.H., S.L. and V.D. All the authors contributed to evaluation of the modelling, discussion of the results and the writing of the paper.

## Competing interests

The authors declare no competing interests.
