## [Peer Review File · Nature Communications]

Reviewers' comments:

Reviewer #1 (Remarks to the Author):

Unexpected large eruptions from buoyant magma bodies within viscoelastic crust.

By Sigmundsson et al.,

- A review

This paper is well written, topical, and of broad significance to the body of knowledge in the fields of volcanotectonics and volcano seismology. The manuscript presents a new model for how fractures in tension might interact with deep magma chambers to reveal how large eruptions may occur with lower-than-expected levels of warning. I have only some very minor comments and suggestions (below) after which the manuscript is suitable for publication in Nature Communications.

Minor comments:

- 1) Line 30: suggest "...magma body may decrease well..."
- 2) Line 70: I'd suggest "neglected" rather than "ignored".
- 3) Line 165: Again, I'd suggest use "neglected" rather than ignored.

A couple of more general points, for consideration:

- Might the author write a line or two explaining how the model (whether spherical or elliptical) is reasonable given the likely complexities in the magma chamber? I fully realize that the proposed model dramatically simplifies the calculations, and in fact I agree that it is fine, but with the community now starting to see the inherent limits of the so-called "Mogi model" I wonder if the authors might comment on some of the limitations?
- Figure 3 & 4, and also lines 151-154 & 160-165: might authors again comment (perhaps in the discussion) as to what the likely "stable geometry" of the "sustained channel" might be? This is especially important if one considers the local stresses, which might evolve from one set of conditions below the brittle-ductile level (and the deep chamber) to another set of conditions supporting the lateral dyke.

Philip Benson, Portsmouth/London, October 2019

Reviewer #2 (Remarks to the Author):

Dear Authors,

I have been reading with interest your contribution. The simple analysis you use is interesting and it is in my opinion contributing to stress that eruptions triggered by buoyancy could be preceded by little warning, which is very important for volcanic hazard assessment.

I find the manuscript interesting, even if portions of it can be improved for clarity. I provided some suggestions in the annotated version, but I am not a native English speaker and I leave it to authors with better English skills than mine to clean up the text.

I have to say that Figure 3 and 4 are not of the quality required for publication in Nature Communications. Figure 3 should not be schematic but rather show the influence of the different parameters of Equation 4 on the evolution of pressure during eruption. Figure 4 can be a summary figure but should be significantly improved. I know it is not an art context, but still...

I have one major comment: What about volatiles?

The Bardarbunga-Holuhraun magmas contained quite a lot of excess fluids. What is the impact of excess volatiles on the evolution of overpressure in the magma reservoir and on magma ascent within the dyke?

1. The presence of excess volatiles in the reservoir that fed the eruption will increase magma compressibility, decreasing even further the overpressure generated by magma injection (see Kilbride et al., 2016, Nat.Comm).
2. Excess fluids at depth will decrease magma density and thus increase buoyancy
3. Can the progressive expansion of excess fluids help maintaining the dyke open?

I think these aspects should be discussed quite in depth in the main text as they could reinforce your conclusions and otherwise bring quite some criticisms from readers.

Reviewers' comments:

Reviewer #1 (Remarks to the Author):

Unexpected large eruptions from buoyant magma bodies within viscoelastic crust.

By Sigmundsson et al.,

- A review

This paper is well written, topical, and of broad significance to the body of knowledge in the fields of volcanotectonics and volcano seismology. The manuscript presents a new model for how fractures in tension might interact with deep magma chambers to reveal how large eruptions may occur with lower-than-expected levels of warning. I have only some very minor comments and suggestions (below) after which the manuscript is suitable for publication in Nature Communications.

Minor comments:

- 1) Line 30: suggest "...magma body may decrease well..."
- 2) Line 70: I'd suggest "neglected" rather than "ignored".
- 3) Line 165: Again, I'd suggest use "neglected" rather than ignored.

These suggested changes have been made.

A couple of more general points, for consideration:

- Might the author write a line or two explaining how the model (whether spherical or elliptical) is reasonable given the likely complexities in the magma chamber? I fully realize that the proposed model dramatically simplifies the calculations, and in fact I agree that it is fine, but with the community now starting to see the inherent limits of the so-called "Mogi model" I wonder if the authors might comment on some of the limitations?

We have added sentences at the beginning of the paragraph with equation (4) to further justify the use of an elliptical model and put our simplifications into context of models of magmatic systems. New references in these sentences refer to the recent overview paper by Cashman et al. (2018) on this topic, and Sigmundsson (2016, 2019) for the use of the concept of a magma domain introduced in the new sentences. These new references are number 34, 37 and 38.

- Figure 3 & 4, and also lines 151-154 & 160-165: might authors again comment (perhaps in the discussion) as to what the likely "stable geometry" of the "sustained channel" might be? This is especially important if one considers the local stresses, which might evolve from one set of conditions below the brittle-ductile level (and the deep chamber) to another set of conditions supporting the lateral dyke.

Philip Benson, Portsmouth/London, October 2019

We now comment on this in paragraph 4 in the chapter on “Application to Bárðarbunga volcano”. There we say “ We suggest a vertical upflow path along the eastern caldera boundary of Bárðarbunga (Supplementary Text). This initial flow path from the magma body may have been a dike that formed along a part of the caldera boundary where it was weak. At a higher level, within the brittle crust, where the magma is less buoyant and topographic and tectonic stress have major influence, a regional dike formed (Fig. 7).”.

Reviewer #2 (Remarks to the Author):

Dear Authors,

I have been reading with interest your contribution. The simple analysis you use is interesting and it is in my opinion contributing to stress that eruptions triggered by buoyancy could be preceded by little warning, which is very important for volcanic hazard assessment.

I find the manuscript interesting, even if portions of it can be improved for clarity. I provided some suggestions in the annotated version, but I am not a native English speaker and I leave it to authors with better English skills than mine to clean up the text.

All the changes suggested in the annotated version have been considered and most of them implemented. Critical reading by native English speakers for grammar/sentence structure has been done, and text improved throughout.

I have to say that Figure 3 and 4 are not of the quality required for publication in Nature Communications. Figure 3 should not be schematic but rather show the influence of the different parameters of Equation 4 on the evolution of pressure during eruption. Figure 4 can be a summary figure but should be significantly improved. I know it is not an art context, but still...

Original figures 3 and 4 have been revised and improved. In the revised manuscript they are figures number 5 and 7 (after moving material from the supplementary material into the main body of the text). The new figure 5 explains better the effects of the height of the sustained magma channel and density difference, and also shows effects of volatile release. The influence of the value of different parameters is now evaluated in a new paragraph at the end of the section on “Application to Bárðarbunga volcano”.

I have one major comment: What about volatiles?

The Bardarbunga-Holuhraun magmas contained quite a lot of excess fluids. What is the impact of excess volatiles on the evolution of overpressure in the magma reservoir and on magma ascent within the dyke?

1. The presence of excess volatiles in the reservoir that fed the eruption will increase magma compressibility, decreasing even further the overpressure generated by magma injection (see Kilbride et al., 2016, Nat.Comm).
2. Excess fluids at depth will decrease magma density and thus increase buoyancy
3. Can the progressive expansion of excess fluids help maintaining the dyke open?

I think these aspects should be discussed quite in depth in the main text as they could reinforce your conclusions and otherwise bring quite some criticisms from readers.

We have followed this suggestion. The main changes in the new version of the manuscript include:

- (i) a new paragraph at the end of chapter on the “Modelling framework” describing the general effects of volatiles.
- (ii) a specific evaluation of the volatiles in the case of the Bardarbunga events, in the new first paragraph in the section on “Application to Bárðarbunga volcano”.
- (iii) new Figure 6 addressing the effects of volatiles.
- (iv) a new paragraph in the Discussion on the effects of volatiles

Significant comment from annotated manuscript other than related to presentation:

What about the presence of excess fluids during magma ascent. These magmas contain significant amount of CO₂ and will therefore be rather compressible. Wouldn't the progressive exsolution of volatiles during ascent contribute to help keeping the dyke open? If not, it should still be explained in the text how the presence of excess volatiles (which are surely there) will be on magma ascent and the capacity of the dyke to remain open.

We now address the progressive exsolution of volatiles as explained above. The specific question if exsolution of volatiles help keep the dyke open is addressed at the end of the new volatile paragraph in “Discussion”. Although this is important in the top few km of the crust (above 3 km), our evaluation suggests it is actually not an important factor in keeping a dike open at large depth (below 3 km).

I do not find Figure 3 particularly insightful. Instead of a schematic figure I would prefer to see calculations performed with Equation 4 and showing the impact of the variability of the different parameters on the resulting decrease of pressure in time.

See earlier response to comment on Figure 3.

what does sufficiently large mean [for triggering a caldera collapse]? Specify a value or a stress distribution that is compatible with caldera collapse. This is not obvious to non-specialists.

We modified the text here so it reads “...Depressurization in the model produces strong shear stress along potential ring faults location at depth, which favors the initiation a piston-type collapse (Fig. 8); large under-pressure allowed for the onset of movements on already weak caldera faults following previous events (Supplementary Text) ...”

We thus now describe the stress distribution, which is compatible with caldera collapse. Our previous statement was inaccurate as it is actually uncertain what is a sufficiently large stress to create a caldera collapse. That may vary from one volcano to another, but can be expected to be lower where there are pre-existing caldera faults compared to volcanoes where no collapse has occurred.

Other comments in annotated manuscript (mostly minor, or related to presentation)

All these other comments/suggestions have been considered and addressed when revising the paper.